# Numerical Simulation and Experimental Verification of Droplet Generation in Microfluidic Digital PCR Chip

**DOI:** 10.3390/mi12040409

**Published:** 2021-04-07

**Authors:** Xiangkai Meng, Yuanhua Yu, Guangyong Jin

**Affiliations:** 1School of Life Science and Technology, Changchun University of Science and Technology, Changchun 130022, China; mxk2018@cust.edu.cn (X.M.); Yhycust@163.com (Y.Y.); 2School of Science, Changchun University of Science and Technology, Changchun 130022, China; 3Key Laboratory of Biological Detection Engineering, Changchun 130022, China

**Keywords:** droplet digital PCR, droplet generation, microfluidic, COMSOL Multiphysics

## Abstract

The generation of droplets is one of the most critical steps in the droplet digital polymerase chain reaction (ddPCR) procedure. In this study, the mechanism of droplet formation in microchannel structure and factors affecting droplet formation were studied. The physical field of laminar two-phase flow level was used to simulate the process of droplet generation through microfluidic technology. The effect of the parameters including flow rate, surface tension, and viscosity on the generated droplet size were evaluated by the simulation. After that, the microfluidic chip that has the same dimension as the simulation was then, fabricated and evaluated. The chip was made by conventional SU-8 photolithography and injection molding. The accuracy of the simulation was validated by comparing the generated droplets in the real scenario with the simulation result. The relative error (RE) between experimentally measured droplet diameter and simulation results under different flow rate, viscosity, surface tension and contact angle was found less than 3.5%, 1.8%, 1.4%, and 1.2%, respectively. Besides, the coefficient of variation (CV) of the droplet diameter was less than 1%, which indicates the experimental droplet generation was of high stability and reliability. This study provides not only fundamental information for the design and experiment of droplet generation by microfluidic technology but also a reliable and efficient investigation method in the ddPCR field.

## 1. Introduction

The definition of digital polymerase chain reaction (dPCR) was proposed by B.Vogelstein in 1990, and a mutated oncogene was detected in the feces of patients with colorectal cancer through this method [1]. In digital PCR, the DNA or RNA is separated into numerous similar volumes of droplets, in which molecules are randomly distributed. The fluorescence could then digitally read and count after the amplification procedure [2,3]. In the past decades, droplet digital polymerase chain reaction (ddPCR) based on the microfluidic chip has been widely used in a variety of applications as the replacement to the conventional PCR techniques [4,5], due to the advantage of high sensitivity, specificity, and accuracy [6], ddPCR does not rely on cycle threshold (CT) value for quantitative analysis and has good repeatability [7]. It is especially suitable for detecting rare gene mutation, subtle copy number variation, and absolute quantitative analysis of nucleic acid [8].

The droplet generation is the most critical step of the ddPCR procedure. The digital droplet technology based on microfluidics can quickly prepare droplet PCR reaction units of uniform size, which is an ideal digital PCR platform [9,10]. The quality of droplet generation determines the accuracy of subsequent PCR and fluorescence detection [11]. With the same PCR reagent consumption, smaller generated droplet size could, to some extent, indicate the larger number of microreactors and hence archive the higher detection sensitivity. Vice versa, too small droplet size and too large amount of droplet are likely to increase the detection cost, prolong the detection time, and decrease the limit of detection. Therefore, understanding the droplet generation system and precisely predict and balance the size of the generated droplet is crucial for the entire ddPCR processing. The influencing factors of droplet diameter in the microfluidic chip should be studied. This not only greatly simplifies the design process of the microfluidic chip, but also greatly improves the reliability of the designed microchannel structure. Recently, several works were reported concerning the droplet formation process and related factors in microchannel through experimental analysis and numerical simulation. Anna et al. [12] and Dreyfus et al. [13] first proposed a microfluidic focusing chip to generate droplets and found that the size of the generated droplets is not only related to the flow rate ratio of the two phases but also related to the size of the microchannel. Garstecki et al. [14] used a cross-focus microfluidic device to generate monodisperse bubbles. Takeuchi et al. [15] applied the circular channel to the three-dimensional flow focusing model. Compared with the two-dimensional flow focusing model, it improved the flow rate and the yield of droplets. At the same time, it also improved the sensitivity of the micro-droplet diameter to fluid parameters. Fu et al. [16] experimentally studied the dynamics of droplet formation and rupture in the drop or ejection state in a flow-focused microchannel. Zhou [17] proposed that the mechanism of generating droplets by pinching discrete phases applied to both flow patterns. Li et al. [18] used the lattice Boltzmann method to numerically simulate the formation of droplets of the liquid two-phase flow in the cross-focused microchannel and verified the accuracy of the simulation through experiments. Sur et al. [19] used the volume of fluid (VOF) method to simulate the gas-liquid two-phase flow in the cross-focused microchannel and studied the flow patterns under different inertial forces, viscous shear forces, and surface tensions.

However, the currently published research mainly focuses on the studies of droplet formation. Numerical studies on the different influence factors of droplet diameter using microfluidic technology are rarely reported, especially the formation of stable droplets formed by microfluidic digital PCR chip. Therefore, the numerical simulation and experimental verification of droplet generation in microfluidic digital PCR chips have very important scientific and practical values. In this paper, numerical simulation and experimental verification are performed by COMSOL Multiphysics 5.4 to study the generation mechanism and influencing factors of droplet diameter. The crucial factors influencing the diameter of generated droplets are studied, including the flow rate ratio of two-phase flow, the viscosity of continuous phase, the surface tension, and the contact angle. The research on droplet generation in a flow focusing structure provides guidance for practical applications of droplet digital PCR chips.

## 2. Materials and Methods

### 2.1. Materials

Photoresist SU-8 (Micro Chem 2050), copolymers of cycloolefin (COC)(TOPAS 5031), (tridecafluoro-1,1,2,2 tetrahydrooctyl) trichlorosilane (United Chemical Technologies, Inc., Bristol, PA), an inverted microscope (IX73, Olympus Corp.) was used to observe and image the experiments, Manta G-1236 CMOS camera, UV light, silicon wafer, tabletop homogenizer, incubator, plasma cleaning machine, air pumps, proportional valve, solenoid valve, gas cylinder, silicon oils (5 cst, 20 cst, 60 cst, Dow Corning, USA) with different viscosity were used as the continuous phase, with 0.05% Triton X-100 (T9284, Sigma Aldrich) and 2% ABIL EM90 supplemented as the surfactant, adjusting the concentration of the surfactant to obtain different surface tensions. De-ionized water was used as the disperse phase. Viscometer (BROOKFIELD DV-II+P, USA) was used to measure the viscosity of liquids at 25 °C, and surface tension tester (Dataphysics DCAT2, Germany) was used to measure the surface tension between oil and water. The droplet generation process was simulated by COMSOL Multiphysics 5.4, which has a rich mathematical model library and possesses a powerful post-processing function of simulation results, and multi-physics can be accurately and effectively described through partial differential equations. Because of the finite element method, the calculation accuracy is higher than that of Fluent. The ImageJ software was used to perform image processing on the image obtained in the experiment. After a series of setting such as binarization, edge finding, and threshold adjustment, the diameter of the generated droplets was obtained.

### 2.2. Method

At present, the technology for generating droplets using special-structured microfluidic channels mainly includes the T-shaped interface microchannel, flow focusing microchannel, and coaxial flow microchannel. Among them, the flow focusing microchannel has excellent micro-droplet monodispersity and high yield. At the same time, the diameter of the droplets is easy to control, and is often used for the preparation of droplets in digital PCR [20]. The mechanism of the flow focusing method is mainly capillary instability. The droplet system mainly uses the combined action of fluid shear force and surface tension to obtain droplets. The capillary number (Ca) is the main factor affecting the generation of droplets [21]. In the microfluidic focusing device, droplets are generated due to the instability of the capillary. The mathematical expression of the capillary number (Ca) is:(1)$$\mathrm{Ca}=\frac{\mu_{c}v_{c}}{\sigma },$$
where$\;\mu_{c}$ is the viscosity of the continuous phase. $v_{c}$ stands for continuous phase flow rate. σ represents the two-phase surface tension. Therefore, to meet the need of microfluidic dPCR chip, the diameter of the droplet and stability are controlled by changing the capillary numbers (Ca). That means to change the continuous phase flow rate, the viscosity of the continuous phase, and surface tension. Therefore, corresponding numerical simulation and experimental research are carried out for the flow focusing structure.

Figure 1 displays the research route of droplet generation and validation experiment. The numerical simulation was established, and the ddPCR droplet generation chip was fabricated by soft lithography. A droplet generation device was built in the laboratory to provide two-phase flow pressure.

#### 2.2.1. The Function of Numerical Simulation

In the droplet motion simulation, the following assumptions were established: the liquid phase material was an incompressible Newtonian fluid. In addition, the fluid characteristics were steady and did not change with time. The pressure drive was stable and did not change with time. The droplet motion model was described in this paper can be conveniently set up through the level set interface of laminar two-phase flow in COMSOL Multiphysics 5.4. Navier Stokes Equation (2) and continuity Equation (3) can be established in the laminar flow interface. In the level set interface, the level set Equation (4) can be selected for level set variables [22]:(2)$$\rho \frac{\partial u}{\partial t}+\rho \left(u\cdot \nabla \right)u=\nabla \cdot \left[-pI+\mu \left(\nabla u+{\left(\nabla u\right)}^{T}\right)\right]+F_{\mathrm{st}},$$
(3)∇·u=0,
(4)$$\frac{\partial \phi }{\partial t}+u\cdot \nabla \phi =\gamma \nabla \cdot \left(\varepsilon \nabla \phi -\phi \left(1-\phi \right)\frac{\nabla \phi }{\left|\nabla \phi \right|}\right).$$

In the above equation, ρ is the density (kg/m^3^), u is the velocity (m/s), *t* is the time (s), μ is the dynamic viscosity (Pa**·**s), p is the pressure (Pa), $F_{\mathrm{st}}$ is the surface tension (N/m^3^). Besides, ϕ is the level set function, and the interface of two-phase flow is defined by ϕ = 0.5, namely the droplet surface (the dispersed phase is at ϕ = 0 and the continuous phase is at ϕ = 1), γ and ε are numerical stability parameters. The density ρ and dynamic viscosity coefficient μ at the interface can be determined by Equations (5) and (6) [23]:(5)$$\rho =\rho_{1}+\left(\rho_{2}-\rho_{1}\right)\phi ,$$
(6)$$\mu =\mu_{1}+\left(\mu_{2}-\mu_{1}\right)\phi ,$$
where $\;\rho_{1}$,$\;\rho_{2}$,$\;\mu_{1}$, and $\mu_{2}\;$ represent the density and viscosity of fluid 1 and fluid 2, respectively. To calculate the effective droplet diameter $d_{eff},\;$ an integration operator was used to find the area corresponding to the dispersed phase, as shown in Formula (7):(7)$$d_{eff}=2\cdot \sqrt[{3}]{\frac{3}{4\pi }\int_{\Omega }^{\;}\left(\phi >0.5\right)d\Omega }.$$

Here, Ω represents the leftmost part of the horizontal channel, where x < −0.2 mm and ϕ>0.5, as shown in Figure 2a.

#### 2.2.2. Building Geometric Model

Figure 2b shows the three-dimensional structure model of a flow focus microchannel with a square cross-section, and dimension of 80 μm × 80 μm. It was enough to model half of the flow focus microchannel geometry because of the symmetry. The sample reaction system (fluid 2), which was dispersed into small droplets, flowed from right to left through a horizontal channel (inlet 1). In addition, the continuous phase flow (fluid 1) entered through a vertical channel from two directions (inlet 2 and inlet 3).

#### 2.2.3. Simulation of the Microstructure of Chip

In this study, the flow focusing structure was investigated, and the simulation model of the structure was built. Firstly, the physical field of the laminar two-phase flow level set was added, then the geometric model was built, and the material was added. Then, boundary conditions and meshing were established, the results displayed, and post-processing was finished after process solving.

#### 2.2.4. Initial Boundary Conditions

The droplets are generated by two-phase flow, which the silicon oil (5 cst) as a continuous phase and the sample (de-ionized water) as a discrete phase. According to the Formula (1), the diameter of droplet in micro-channels is mainly affected by viscous force and surface tension. The physical properties of the two-phase flow which are related to droplet formation diameter are shown in Table 1.

Assuming that the channel was completely filled with silicon oil at the initial time of flow, starting from time *t* = 0, oil and water flow in from the two-phase inlet at fixed flow rates V_c_ (flow rate of continuous phase) and V_d_ (flow rate of discrete phase), respectively, as shown in Table 2. At the three inlets, the laminar inflow condition of the specified volume flow was used. The continuous phase filled in the microstructure in the initial state, and with the flow rate V_c_ from the inlet 1 and inlet 2 injected to the microchannel. At the outflow boundary, set the condition of pressure and no viscous stress. The sample with flow rate V_d_ injected into the microstructure from inlet 3. Additionally, the effective droplet diameter $d_{eff}$ was calculated by the Formula (7). The wetting wall multi-physical field boundary condition applied to the contact angle of 135 degrees.

The contact angle θ was 3/4π, which was the angle between the fluid interface and the wall at the contact point between the fluid interface and the solid wall. In addition, the region where x < −0.2 mm was the effective droplet generation part. The surface tension coefficient was set to 5 × 10^−3^ N/m. The slip length β was the distance to the position where the extrapolated tangential velocity component was zero outside the wall, as shown in Figure 3. All solid boundaries with slip length equal to mesh size parameter.

#### 2.2.5. Fabrication of Chips

The chip structure was designed and fabricated based on the simulation result. The microfluidic chip was fabricated by conventional photolithography [24,25,26,27,28,29] and COC injection mold. The microfluidic device design is shown in the left side of Figure 4.

The device master mold is fabricated by conventional SU8 photolithography, after development, the master mold was hard-baked at 150 °C for 30 min and followed by a salinization with (tridecafluoro-1,1,2,2 tetrahydrooctyl) trichlorosilane (United Chemical Technologies, Inc., Bristol, PA) for 5 min to preventing pattern removal during the polymer replication process. Then, the COC particles were put into the injection molding machine (CT80M8), the mold temperature was set to 60 ℃, and the melt temperature was set to 215 ℃, injection speed was set to 97 mm/s, the injection pressure was set to 60 MPa, and the holding pressure was set to 60 MPa. The molten COC entered the mold under the push of the screw. After cooling, the chip structure layer with a microchannel structure was obtained. The required COC chip could be obtained by hot pressing and bonding the chip structure layer with another COC plastic sheet after oxygen plasma treatment (PDC-MG, Harrick Plasma) for 1 min and baked for 6 min. Holes with diameters of 1 mm were punched using biopsy punches.

## 3. Simulation and Experimental Results

### 3.1. Simulation Test and Mesh Refinement Assay

After setting the initial conditions, the mesh was established. Additionally, the transient solution domain was set, then the calculation was carried out, at last, the results were obtained and post-processed. The fluid interface where the level set function ϕ = 0.5 and the velocity streamline were displayed at different times. The simulation droplet generation process is demonstrated in Figure 5a. To improve the accurate simulation results, a mesh independence assay was conducted. The cross channels were divided by regular hexahedral mesh. The relationship between the diameter of the generated droplets and the number of elements in mesh is shown in Figure 5b. The simulated value converges when the elements is equal or greater than 60,000. It can be considered that the numerical simulation with the number of elements in mesh of 60,000 is a mesh independent solution. 60,000 elements were then used for the simulation of this work.

### 3.2. The Effect of Flow Rate Ratio, Continuous Phase Viscosity, Surface Tension, and Contact Angle on the Diameter of Droplets

The effective droplet diameter $d_{eff}$ is the diameter of a spherical droplet with the same volume as the droplet formed and is calculated by the Equation (7). After determining the size of the microchannel and fluid parameters, the flow rate is a key factor affecting the diameter of the droplet. The effect on the droplet diameter was simulated in COMSOL Multiphysics 5.4 by adjusting the value of V_d_ (water) and V_c_ (oil), and other parameters remain unchanged in Table 1 and Table 2 and the relationship between the effective droplet diameter and six different two-phase flow rate ratios is shown in Figure 6a.

The simulation result shows that at a fixed flow rate of the discrete phase, as the flow rate of the continuous phase increases, when the flow rate ratio (V_d_/V_c_) increases from 1: 8 to 4:1, the effective diameter of the droplets increases from 87.69 to 127.23 μm (Figure 6a), increased by 45.1%, this is because the squeezing force and viscous tension of the continuous phase on the discrete phase at the intersection of the microchannel increase, the expansion process of the discrete phase becomes shorter, and the size of the generated droplets changes smaller, and with the increase of the continuous phase flow rate, the two-phase flow rate has a smaller and smaller influence on the droplet diameter. And a polynomial fitting is carried out for the influence on the droplet diameter under different flow rate ratio, the result is as follows: y = 88.62 + 21.11x − 2.93x^2^, where y is the droplet diameter and x is the flow rate ratio. Compared with the experimentally measured droplet diameter, the relative error was less than 3.5%.

In this part of the simulation, the effects of continuous phase viscosity, surface tension, and contact angle on droplet size were studied separately by numerical simulation. Figure 6b shows the relationship between the diameter of droplet generation and the viscosity of the continuous phase. When other parameters unchanged in Table 1 and Table 2, the viscosity of the continuous phase increases from 1 × 10^−^^2^ Pa∙s to 9 × 10^−^^2^ Pa∙s under simulation conditions. In actual experiments, silicone oils (5 cst, 20 cst, and 60 cst) with viscosity of 1.12 × 10^−2^ Pa·s, 2.68 × 10^−2^ Pa·s and 7.25 × 10^−2^ Pa·s were used as the continuous phase. As the viscosity of the continuous phase increases, the diameter of the effective droplets decreases from 110.25 μm to 100.03 μm (Figure 6b), 9.2% reduction. This is because as the viscosity of the continuous phase becomes higher, the viscous force becomes larger, thereby accelerating the process of entrainment focusing, and thus shortening the formation time of droplets. Therefore, changing the continuous phase viscosity is also an effective method to control the droplet diameter. We further studied the polynomial fitting and got the following results: y = 111.13 − 0.59x − 0.07x^2^, where y is the droplet diameter and x is the continuous phase viscosity. The maximum relative error between the experimentally measured droplet diameter and simulation results is 1.8%.

The effect of surface tension to the generated droplet size is illustrated in Figure 6c. When other parameters remain unchanged in Table 1 and Table 2, the surface tension increases from 5 × 10^−3^ N/m to 25 × 10^−3^ N/m under simulation conditions. In the experiment, the surface tension between oil and water was changed by adjusting the concentration of surfactant, and five experimental conditions with surface tensions of 4.5 × 10^−3^ N/m, 9.6 × 10^−3^ N/m, 14.3 × 10^−3^ N/m, 19.8 × 10^−3^ N/m, and 23.9 × 10^−3^ N/m were obtained. When the surface tension increases, the size of the droplets increases from 105.72 μm to 114.41 μm (Figure 6c), increased by 8.2%. This is because the surface tension belongs to the interaction between the two phases, when the surface tension increases, it will overcome the shear of the viscous force on the dispersed phase and tend to shrink inward. Therefore, as the surface tension increases, the time for flow focusing also increases. This means that the effective diameter of the droplet increases. The influence on the droplet diameter under different surface tension was produced by polynomial fitting, and we obtain the following results: y = 101.48 + 0.72x − 0.01x^2^, where y is the droplet diameter and x is the surface tension. The experiments show that the relative error is less than 1.4%.

In addition, the hydrophobicity of the microchannel surface determines the wettability, and hence affects the droplet formation process. The nature of the interaction between the droplet and the channel wall depends on the contact angle. In the numerical simulation, the wettability of the microchannel wall can be studied by changing the contact angle. To study the influence of microscopic properties on micro-scale generation, this section uses five different contact angles of 120°, 135°, 150°, 165°, and 180° to simulate the generation of the droplet, and the other parameters remain unchanged in Table 1 and Table 2. In the experiment, the hydrophobicity of the chip surface was modified, and four different contact angles of 115°, 132°, 145°, and 163° were obtained after UV treatment under different exposure times. As shown in Figure 6d, when the contact angle increases from 120° to 180°, the diameter gradually decreases from 104.96 μm to 96.62 μm (Figure 6d), 7.6% reduction. This is because the decrease of the adhesion on the surface will reduce the flow resistance, thereby reducing the time to form a droplet at the position of the flow-focus structure and the time to break from the dispersed phase. The relation between the contact angle and droplet diameter is depicted by means of the linear fitting, the relation expression is y = 121.34 − 0.14x, where y is the droplet diameter and x is the contact angle. It is found that the relative error is less than 1.2% by experimentally measuring the droplet diameter under different contact angles.

### 3.3. Fabricated Device and Experimental Setup for Validation

The droplet generation system was used to generate the stable pressure required for droplet formation. Its main components include an air pump, two pressure sensors, a gas cylinder, a pressure control module, and several solenoid valves, as shown in Figure 7a. The system could produce two highly stable air pressures by proportion integral differential (PID) closed-loop control and the fluctuation range of droplet formation was only ±0.1 mbar. The chip clamps were used to fix the droplet generation chips, and the air pressure was adjustable by the potentiometer. When the instrument worked continuously for 2 h, and the fluctuation range was ±1 mbar (0.1%). The dimension of the fabricated channels possessed good consistency, with a measured cross-section of 82.3 μm × 79.1 μm and relatively smooth surfaces. The droplet generation chip was fabricated by COC injection molding [30,31] and its microstructure was observed through a microscope, as shown in Figure 7b,c.

After injecting the sample and oil into the inlet and setting the pressure, the microchannel cross-section was 80 μm × 80 μm, and the flow rate (V_d_/V_c_) was changed from 1:8 to 4:1, and the droplets were collected in the collecting cavity. After that, the droplets were transferred by pipette and tiled on the slide. Then, the slide was put under a microscopic imaging device. The droplet diameter was measured by the ImageJ software, the distance (D1) of five droplets was measured, and the average diameter was calculated. The results are shown in Figure 6a and Table 3.

It can be seen from the result in Figure 6a and Table 3, the experimentally measured droplet diameter under different flow rate ratios is in good agreement with the simulation results, relative error was less than 3.5% (V_d_/V_c_ = 1:8). Statistically, the *p* values of one-way ANOVA were calculated and ranged from 0.75 to 0.83. From this, it can be concluded the data collected from both methodologies were not statistically different from one another at a confidence level of 95%. It proves that the numerical simulation has guiding significance for the actual experiment, and the diameter of droplet can be accurately predicted and controlled according to the numerical simulation.

To verify the actual effect of other parameters including viscosity of continuous phase, surface tension, and contact angle on droplet formation, this paper verifies the three parameters experimentally and compares them with the previous numerical simulation results. The comparison results are shown in Figure 6. The viscosity of continuous phase, surface tension, and contact angle are changed by using different silicon oil, changing the concentration of the surfactant, and the surface hydrophobic treatment of the chip material. As shown in Figure 6b, when other parameters unchanged in Table 1 and Table 2, the experimentally measured droplet diameter under three different viscosity of the continuous phase (1.12 × 10^−2^ Pa·s, 2.68 × 10^−2^ Pa·s and 7.25 × 10^−2^ Pa·s) is in good agreement with the simulation results, relative error was less than 1.8%. As shown in Figure 6c, when other parameters unchanged in Table 1 and Table 2, the experimentally measured droplet diameter under five different surface tension (4.5 × 10^−3^ N/m, 9.6 × 10^−3^ N/m, 14.3 × 10^−3^ N/m, 19.8 × 10^−3^ N/m, 23.9 × 10^−3^ N/m) is in good agreement with the simulation results, relative error was less than 1.4%. As shown in Figure 6d, when other parameters were unchanged in Table 1 and Table 2, the experimentally measured droplet diameter under four different contact angles (115°, 132°, 145°, 163°) is in good agreement with the simulation results, relative error was less than 1.2%. There is no significant difference.

The uniformity of the generated droplet is then evaluated by visual measurement of the length of five droplets using Image J, three droplet generation chips were tested under V_d_/V_c_ (2:1), other parameters unchanged in Table 1, the droplet diameter was measured, and the results are shown in Table 4. The average diameter was 115.28 μm, not much difference from simulation results. The standard deviation of the average diameter was 1.04 μm and the coefficient of variation was about 0.90%, which proved good homogeneity of droplets generation. It could fully meet the needs of biochemical test detection and analysis.

## 4. Conclusions

In this paper, the process of droplet generation in the droplet generation chip was simulated by level set interface of laminar two-phase flow using COMSOL. The parameters of flow rate ratio, continuous phase viscosity, surface tension, and contact angle were evaluated in a real scenario. Subsequently, the influence of these parameters on the diameter of micro-droplets was studied through experiment. The relative error between the experiments and simulations was respectively less than 3.5%, 1.8%, 1.4%, and 1.2%. The simulation results are in good agreement with the result from visualization measurements. The specific conclusions are as follows.

In the flow focusing microchannel model with a cross-section of 80 μm × 80 μm, the flow rate ratio between the discrete phase and continuous phase has great influence on the size of the generation of micro-droplets: at a fixed flow rate of the discrete phase, the generation size of the micro-droplets decreases with the increase of the continuous phase flow rate. At a fixed flow rate ratio, the diameter of the generated droplets decreases with the increase of viscosity of the continuous phase and increases with the increase of the surface tension. When other conditions are fixed, as the contact angle increases, the droplet diameter decreases.

The uniformity of the droplet diameter was analyzed through experiment, and the coefficient of variation of the droplet size was less than 1%. The uniformity of the droplets was good, which could fully meet the needs of biochemical test detection and analysis in the subsequent ddPCR process. The theoretical and experimental foundation is established for future research on the application of droplets in the digital PCR field. It provides an efficient and reliable study method for the investigation of droplet stable generation and transportation, PCR, and fluorescence detection in the integrated droplet digital PCR system.

## Figures and Tables

**Figure 1 micromachines-12-00409-f001:**
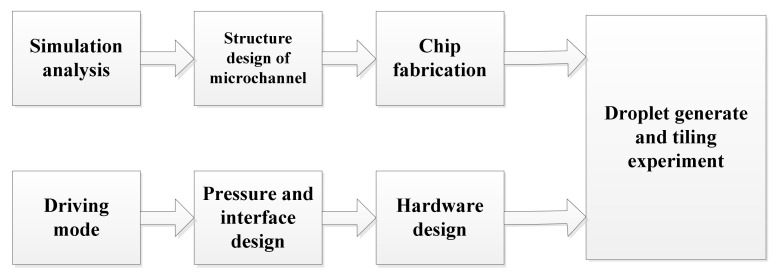
Research route of droplet generation and validation experiment.

**Figure 2 micromachines-12-00409-f002:**
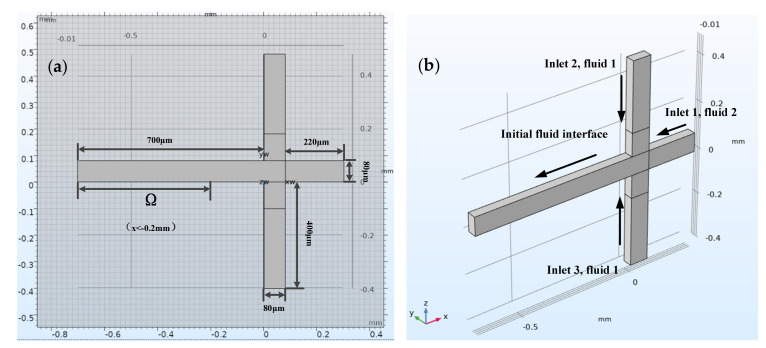
The top view of the flow focusing structure and effective droplet diameter solution area Ω (**a**). 3D model of flow focusing microchannel showing the inflow fluid for each inlet (**b**).

**Figure 3 micromachines-12-00409-f003:**
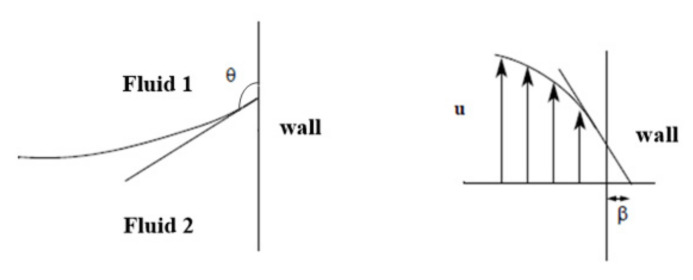
Schematic images showing contact angle θ and slip length β used for this study.

**Figure 4 micromachines-12-00409-f004:**
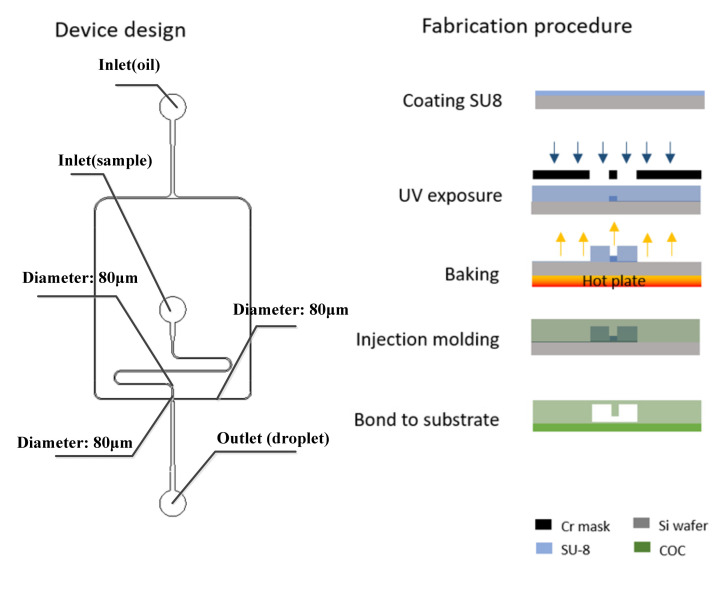
Design (left) and fabrication steps (right) of the microfluidic droplet generator.

**Figure 5 micromachines-12-00409-f005:**
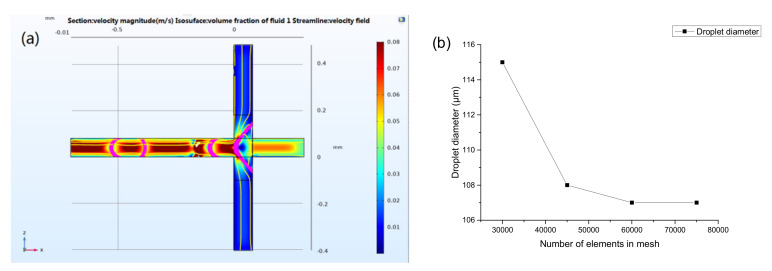
The COMSOL Multiphysics software interface showing a generation of droplet under set boundary conditions (**a**). Mesh refinement assay of the physical model (**b**).

**Figure 6 micromachines-12-00409-f006:**
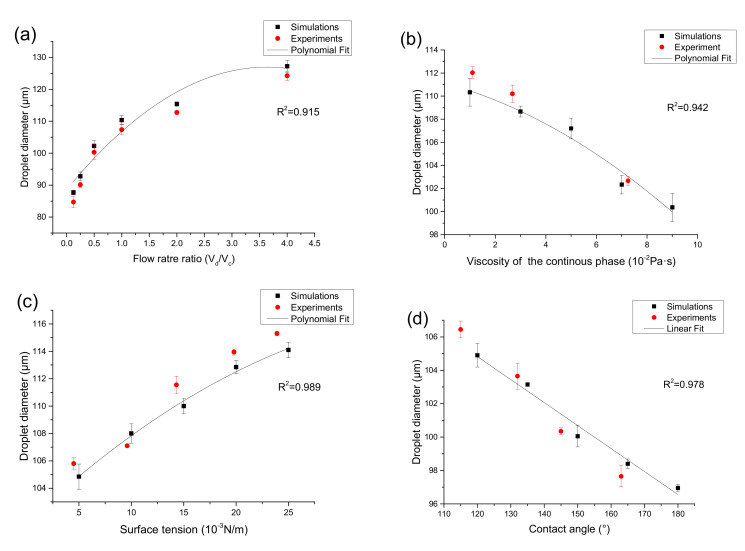
Comparison of simulation droplet diameter with experimental observations under different continuous phase flow rate ratio (**a**), viscosity (**b**), surface tension (**c**), and contact angle (**d**).

**Figure 7 micromachines-12-00409-f007:**
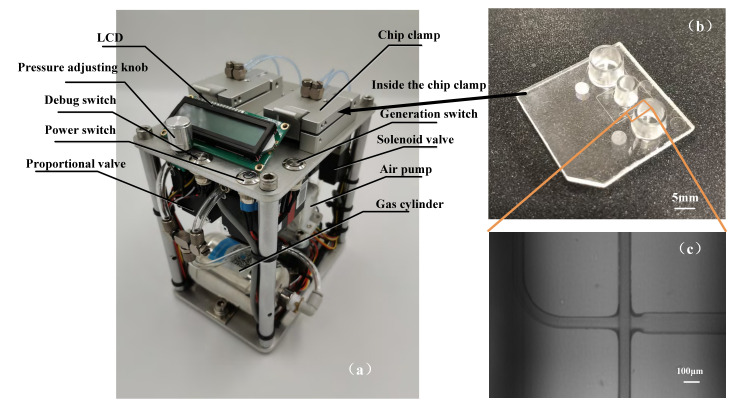
The integrated droplet generation platform for droplet digital polymerase chain reaction (ddPCR) (**a**). The microfluidic droplet generator (**b**). Micrograph Image of actual droplet generation junction (**c**).

**Table 1 micromachines-12-00409-t001:** Physical properties of two-phase flow.

Physical Quantity	Value of Continuous Phase (Fluid 1)	Value of Discrete Phase (Fluid 2)
Density $\left(\mathrm{kg}/m^{3}\right)$	1.164 × 10^3^	1 × 10^3^
Dynamic viscosity (Pa·s)	1.12 × 10^−2^	1.01 × 10^−3^
Surface tension (N/m)	5 × 10^−3^	5 × 10^−3^

**Table 2 micromachines-12-00409-t002:** Definition of condition variables.

Name	Expression	Unit	Describe
V_d_	0.4e − 6/3600 × step 1 (t [1/s]) [m^3/s]	m^3^/s	Flow rate of discrete phase
V_c_	0.4e − 6/3600 × step 1 (t [1/s]) [m^3/s]	m^3^/s	Flow rate of continuous phase
$d_{eff}$	2 × (intop1((phils > 0.5) × (x < −0.2[mm]) × 3/(4 × pi)^(1/3)	m	Effective droplet diameter

**Table 3 micromachines-12-00409-t003:** Simulation and experimental data of droplet diameter under six different flow rate ratios (*N* = 6).

Flow Rate Ratio (V_d_/V_c_)	Simulation Average Droplet Diameter (μm)	Experiment Average Droplet Diameter (μm)	Relative Error (%)
1:8	84.69	87.69	3.5%
1:4	90.09	92.76	3.0%
1:2	99.28	102.20	2.9%
1:1	107.36	110.36	2.8%
2:1	112.74	115.28	2.3%
4:1	124.23	127.23	2.4%

**Table 4 micromachines-12-00409-t004:** Results of the uniformity of the generation droplets (*N* = 9).

Experiment Number	Measured Distance (μm)	Average Diameter (μm)
1-1	571.20	114.24
1-2	568.37	113.68
1-3	572.31	114.46
2-1	575.72	115.15
2-2	577.79	115.56
2-3	576.21	115.24
3-1	582.01	116.40
3-2	580.01	116.00
3-3	584.13	116.83
Average	576.31	115.28
Standard deviation	5.18	1.04
CV%	0.90	0.90

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
