# Peer review of "Numerical Simulation and Experimental Verification of Droplet Generation in Microfluidic Digital PCR Chip"

_micromachines, 2021, doi:10.3390/mi12040409_

Round 1

Reviewer 1 Report

The paper entitled "Numerical simulation and experimental verification of droplet generation in microfluidic digital PCR chip" is about the droplet generation in microfluidic chip. It suffers from critical points :

1- Literature search of dPCR is so weak and the role of the current work on the map is not clear.

2- The novelty of the manuscript is not clear.

3- Numerical part is not justified and formulated well. Properties (physical and numerical) are not mentioned.

4- Expermental philosophy and its design is absent as well as quantatative and qualitative discussion about numerical and experimental agreement.

Reviewer 2 Report

  1. The introduction part is not following a logical flow, there is no literature review concerning the recently published similar works. In the last paragraph of the introduction, the authors need to clarify the advantages of their work, what is the new and better of this work compared with other published works?

The entire structure of the introduction part should be as follows:

  1. Giving a background on what is ddPCR and the advantages of ddPCR (paragraph one)
  2. Why it is important to give good control of generated droplet size and et al. why your research target is significant (paragraph two)
  3. Review the recently published work that using the simulation method to predict the droplet size(paragraph three)
  4. Give a short statement about what you are doing and why it is better than others. (paragraph four)

Minors: Line 49 “How to make sure the diameter of droplets and homogeneity is the crucial point” I would suggest you rephrase the sentence to “How to predict and precise control of the diameter of generated droplets is crucial”. There are a lot of minor changes similar to this, please rephrase and make them more accurately express your idea.

Overall, the authors need to reconstruct the introduction part and give a better logical flow all across it.

  1. Line 77 what is the version of COMSOL software and how the diameter is calculated using ImageJ? please provide more details
  2. Line 87 to line 90, please use the same spacing and font size. This requirement does apply to all the rest of the manuscript, please check carefully before the next submission.
  3. Figure 2 and Table 2 seem not well aligned. please do a proper alignment of all tables and figures before the next submission.
  4. All Figures should be standalone, please add adequate information to figure captions error bars and scales bars.
  5. Table 1 and Line 147 to 148, please provide the reason or reference why these physical properties are chosen
  6. Figure 5, I would suggest you show only 1 3D view for the demo in ESI, show the top view results in the main manuscript as I do not get any useful information from these unclearly presented images. If you want to show the simulation generation of the droplets, use top view, please.
  7. Line 182 to Line 184 “SU8 2050 was spin-coated on a silicon wafer, typically with a thickness of 80 μm, before it was exposed to UV light and developed. After development, the excess photoresist was removed with acetone and dried in the incubator for the 30s to obtain the male mold for chip preparation.” Change to “The device master mold is fabricated by conventional photolithography [reference], after developing, the master mold was hard-baked at XXX ËšC for 30 s before it ready for replica molding”
  8. Line 190, what is the power and treatment time used here for O2 plasma? What kind of biopsy punches did you use to punch the hard plastic material? Any surface salinization need to change the surface from hydrophilic to hydrophobic for the water-in-oil droplet generation?
  9. I can not find a mesh refinement assay, please do the test and add the information to ESI.
  10. Figure 6 should be combined with table 3, show both simulation and real experiment generation at the same place, and also compare the results. Provide a scatter plot with two colors to show the simulated and experimental size of the same set of parameters
  11. Combine figure 8 and figure 9 into one figure. Add scale bar to figure 8, indicate where is figure 8 in figure 9.
  12. Move figure 10 to ESI since it tells barely any information.
  13. Table 3, the title should be “Resulting diameters of droplets measured in three chips”, in addition, what does the “Number” stand for? Also, there is no discussion about how close the simulated size vs experimental droplet size.
  14. Please emphasized the novelty of your work in your conclusion part too.
  15. Overall, I would suggest evaluating more conditions at a different speed, different channel width, and different surface tension. The current data and results seem not completely convincing.

Author Response

Response to Reviewer 2 Comments

Thanks very much for taking your time to review this manuscript. I really appreciate all your comments and suggestions!

Point 1: The introduction part is not following a logical flow, there is no literature review concerning the recently published similar works. In the last paragraph of the introduction, the authors need to clarify the advantages of their work, what is the new and better of this work compared with other published works?

The entire structure of the introduction part should be as follows:

Giving a background on what is ddPCR and the advantages of ddPCR (paragraph one)

Why it is important to give good control of generated droplet size and et al. why your research target is significant (paragraph two)

Review the recently published work that using the simulation method to predict the droplet size(paragraph three)

Give a short statement about what you are doing and why it is better than others. (paragraph four)

Minors: Line 49 “How to make sure the diameter of droplets and homogeneity is the crucial point” I would suggest you rephrase the sentence to “How to predict and precise control of the diameter of generated droplets is crucial”. There are a lot of minor changes similar to this, please rephrase and make them more accurately express your idea.

Overall, the authors need to reconstruct the introduction part and give a better logical flow all across it. 

Response 1: Thank you for your comment. For the introduction part, the logic has been reorganized. First introduced the background and advantages of ddPCR, and secondly explained the purpose of this article in the paragraph two, then reviewed the related research progress. Finally, the innovations of this article are introduced. The minors in Line 49 has been changed (Page 1, Line 48-49)

Point 2: Line 77 what is the version of COMSOL software and how the diameter is calculated using ImageJ? please provide more details.

Response 2: We gratefully appreciate for your valuable comment. The details about the COMSOL software and how the diameter is calculated using ImageJ are provide in the 2.1 Material.(Page 2, Line 96 -99)

Point 3: Line 87 to line 90, please use the same spacing and font size. This requirement does apply to all the rest of the manuscript, please check carefully before the next submission.

Response 3: We apologize for the carelessness. We have changed accordingly and highlighted (Page 3, Line 127 -129)

Point 4: Figure 2 and Table 2 seem not well aligned. please do a proper alignment of all tables and figures before the next submission.

Response 4:  We apologize for the carelessness. We have changed accordingly and highlighted (Page 5, Line187)

Point 5: All Figures should be standalone, please add adequate information to figure captions error bars and scales bars..

Response 5: Thank you for your comment. We have changed accordingly and highlighted ( Figure 6) (Page 8, Line 283)

Point 6: Table 1 and Line 147 to 148, please provide the reason or reference why these physical properties are chosen.

Response 6: Thank you for your comment. The reason for choosing these parameters is according to the equation (1) (Page 3, Line 106-119)

Point 7: Figure 5, I would suggest you show only 1 3D view for the demo in ESI, show the top view results in the main manuscript as I do not get any useful information from these unclearly presented images. If you want to show the simulation generation of the droplets, use top view, please.

Response 7: Thank you for your comment. We have deleted the other 3D view figure, only save Figure 5a to describe the simulation droplet generation process (Page 7, Line228)

Point 8: Line 182 to Line 184 “SU8 2050 was spin-coated on a silicon wafer, typically with a thickness of 80 μm, before it was exposed to UV light and developed. After development, the excess photoresist was removed with acetone and dried in the incubator for the 30s to obtain the male mold for chip preparation.” Change to “The device master mold is fabricated by conventional photolithography [reference], after developing, the master mold was hard-baked at XXX ËšC for 30 s before it ready for replica molding”.

Response 8: Thank you for your comment. We have changed accordingly and highlighted (Page 6, Line 204 -206)

Point 9: Line 190, what is the power and treatment time used here for O2 plasma? What kind of biopsy punches did you use to punch the hard plastic material? Any surface salinization need to change the surface from hydrophilic to hydrophobic for the water-in-oil droplet generation?

Response 9: Thank you for your comment. We have changed accordingly and highlighted (Page 6, Line 204 -213)

Point 10: I can not find a mesh refinement assay, please do the test and add the information to ESI.

Response 10: Thank you for your comment. We have added mesh refinement assay in 3.1 Simulation test and mesh refinement assay and Figure 5b. (Page 6, Line 220 -227)

Point 11: Figure 6 should be combined with table 3, show both simulation and real experiment generation at the same place, and also compare the results. Provide a scatter plot with two colors to show the simulated and experimental size of the same set of parameters

Response 11: Thank you for your comment. We have changed accordingly and highlighted (Page 9, Line 312) Table 3

Point 12: Combine figure 8 and figure 9 into one figure. Add scale bar to figure 8, indicate where is figure 8 in figure 9.

Response 12: Thank you for your comment. We have combine figure 8 and figure 9 into one figure in the Figure 7. The scale bar has been added in Figure 7(b), (c). (Page 9, Line 301)

Point 13: Move figure 10 to ESI since it tells barely any information

Response 12: Thank you for your comment. The Figure 10 has been deleted as there is not enough information.

Point 14: Table 3, the title should be “Resulting diameters of droplets measured in three chips”, in addition, what does the “Number” stand for? Also, there is no discussion about how close the simulated size vs experimental droplet size.

Response 14: Thank you for your comment. We have corrected the title of Table 3(Page 9, Line 311). The “Number” stands for the experiment number of different chip. And the discussion about the simulated size vs experimental droplet size is added in (Page 9, Line 314-320)

Point 15: Please emphasized the novelty of your work in your conclusion part too.

Response 15: Thank you for your comment. We have emphasized the novelty of your work in your conclusion part (Page 10 ,Line 357-370).

Point 16: Overall, I would suggest evaluating more conditions at a different speed, different channel width, and different surface tension. The current data and results seem not completely convincing.

Response 16: Thank you for your comment. We have evaluated more conditions at different surface tension, contact angle, and viscosity of the continuous phase. Figure 6 shows comparison of simulation droplet diameter with experimental observations under different continuous phase flow rate ratio (a), viscosity (b), surface tension (c), and contact angle (d).(Page8 ,Line 283)

Reviewer 3 Report

The authors present an interesting comparison between simulations and experimental results of Droplet Generation in Microfluidic Digital PCR Chip. 

Some points should be clarified before publication.

  • PCR and CT  are acronyms which need to be made explicit the first time you use.
  • The authors should discuss the choice of COMSOL Multiphysics, instead of other simulation methods. A comparison among them should be interessing for the reader. I can suggest some interesting papers:
    1. Balan C M, Broboana D and Balan C 2012 Microfluid. Nanofluid. 13 819
    2. A.Volpe et al. J. Phys. D: Appl. Phys. 50 (2017) 255601
    3. Tsai C H, Lin C H, Fu L M and Chen H C 2012 Biomicrofluidics 6 024108
    4.  Yu Z T F, Lee Y, Wong M and Zohar Y 2005 J. Microelectromech. Syst. 14 1386
  • Line 109: without a picture, the meaning of omega is not clear
  • The subsections of "Materials and Methods" should be numbered. Conversely, it is difficoult for the reader understanding the structure of the paper
  • Figure 6: a table with the exact values could be also provided
  • Line 210: Why  did you test just the flow rate (V1/V2) 2:1? testing all the ratios would have been make your conclusions more reliable.
  • What could change changing the dimensions of the channels? 

Round 2

Reviewer 1 Report

The paper is improved but English writing is poor and the description of pcr chip is missed yet

Reviewer 2 Report

The feedback from the author address my concerns 

I would like to suggest this work be published on Micromachines

However, the grammar and style of the English language need to be improved